# Evaluation of Human Behaviour Detection and Interaction with Information Projection for Drone-Based Night-Time Security

Ryosuke Kakiuchi [1,*], Dinh Tuan Tran [2] and Joo-Ho Lee [2]

1   Guraduate School of Information Science and Engineering, Ritsumeikan University, 1-1-1 Noji-higashi, Kusatsu 525-8577, Japan
2   College of Information Science and Engineering, Ritsumeikan University, 1-1-1 Noji-higashi, Kusatsu 525-8577, Japan; tuan@fc.ritsumei.ac.jp (D.T.T.)
*   Correspondence: kakiuchi.aislab@gmail.com; Tel.: +81-090-4196-0527

**Abstract:** Night security is known for its long hours and heavy tasks. In Japan, a labor shortage of security guards has become an issue in recent years. To solve these problems, an increasing number of robotic security methods are being used. However, several problems exist with existing security robots. For example, wheeled robots traveling on the ground have difficulty in dealing with obstacles such as steps, while most drones are only for monitoring and do not have a function to help people. In this study, an aerial ubiquitous display (AUD) night security drone has been developed to solve the problems of existing security robots. The AUD is equipped with an infrared camera and a projector to detect human behavior at night and present information to people in need. In this paper, an experiment was conducted with the AUD to evaluate whether it can provide adequate nighttime security. In the experiment, real-time monitoring and information projection from the air were achieved. In addition, new security methods using the AUD were shown to be effective. Replacing security guards with the AUD to provide security at night will improve labor shortages in the future, and better security methods will be developed.

**Keywords:** security drone; human–drone interaction; infrared camera; projector

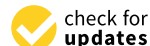



## 1. Introduction

The security industry is known as a heavy and hard-working industry. One example of a job in the security industry is patrolling inside and outside of facilities. They must immediately report suspicious persons and provide accurate information on current locations and destinations to people who are lost or in trouble. These tasks are physically and mentally demanding and sometimes dangerous. In addition, security guards often work at night and for long hours. Furthermore, the number of sites requiring security guards has been increasing in recent years, causing the jobs-to-applicants ratio in the security industry in Japan to increase significantly [1–3]. Figure 1 shows the ratio of effective job offers in the security industry and all jobs in recent years. In addition, security guards incur high labor costs. For this reason, camera-based security methods have become popular in recent years. However, to guard a large area such as a university campus with cameras, a huge number of cameras are required, and it is not possible to provide guidance to people who have lost their way through camera surveillance.

In recent years, security robots such as Guard Robot i [4] and REBORG-Z [5] have been developed to solve this problem. However, all of these security robots are wheeled robots that run on the ground [6,7]. Therefore, there are problems such as the inability to deal with obstacles such as steps, and the difficulty of guarding a wide area at a time. One solution to the above problem is to use drones as security robots. Because drones fly in the air, they are unaffected by obstacles and can monitor a wide area at a time. This makes it possible to quickly and accurately assess the situation. In Japan, ALSOK is using drones to patrol commercial facilities in 2020 [8]. Other research projects in the field of security using

drones include drone surveillance systems for investigation and rescue during natural disasters [9], automated railroad monitoring by drone [10], and real-time detection of violent acts from the air using drones [11]. However, the use of these security drones is limited to brighter areas. In addition, when using a camera mounted on a drone to perform object detection and behavior recognition for security purposes, a main PC is required to process the object detection and behavior recognition. Therefore, it is difficult to provide security using only one drone. This problem requires a robot that can monitor a wide area at night and provide information such as directions to humans with a single robot. In our research, a drone mounted with an infrared camera and a projector is used to solve this problem. The infrared camera is used to recognize human behavior at night, detecting actions such as walking and waving. The projector is used to project information on the ground after approaching a person detected by the infrared camera. The combination of these two technologies enables nighttime security using a security method that has not been possible with previous security drones. In addition, the developed robot is expected to significantly reduce the workload and labor costs of security guards. In addition, since the purpose of this research is for the drone to monitor a wide area from the air at night and provide information to those in need, it is intended for outdoor use only. When monitoring with fixed cameras, it is necessary to install a huge number of cameras to monitor the entire area. The drone in this study solves this problem.

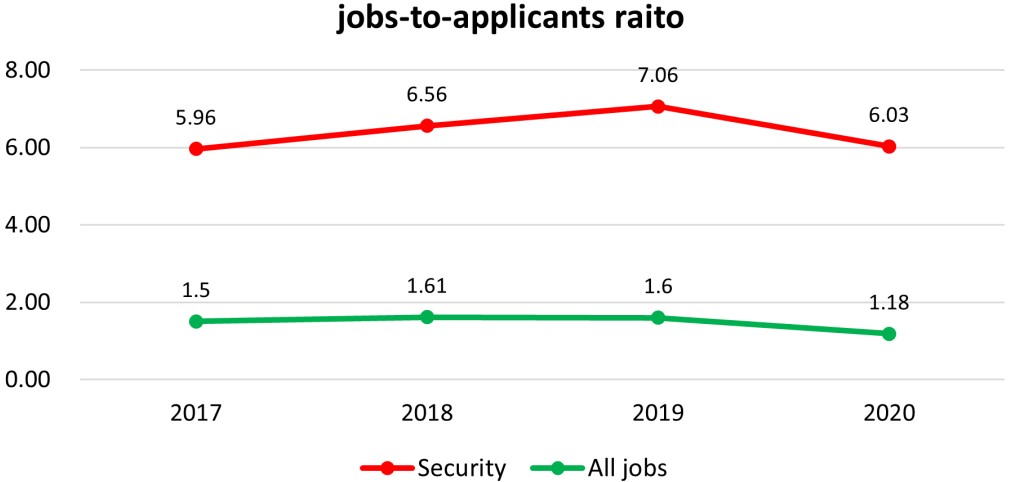

**Figure 1.** Jobs-to-applicants ratio of the security and all jobs.

## 2. Related Works

### 2.1. Creating Datasets for Drone-Based Person Detection

In recent years, drones have become increasingly popular and many studies have been conducted to develop drone-based surveillance systems. Drone surveillance requires the detection of people in the images captured by drones. Mishra et al. [9] have created a dataset for object detection and action recognition by drones. Multiple people performing various actions were filmed from the air at various altitudes and angles using a camera mounted on a drone. A dataset of people walking and waving was created by annotating these images. However, no action detection model has been created and no action detection has been carried out using the video images taken from the air. We have created a dataset and an inference model using aerial images taken with an infrared camera. Running the inference model on a small PC on a drone enables real-time action detection from the air.

### 2.2. Person Recognition Using Aerial Video

Iwashita et al. [12] detected objects in aerial video and classified them based on their movements. By analyzing the trajectory of each object, it was determined whether the moving object was a person or noise. A Kalman filter was used to analyze the trajectories.

The object trajectories were calculated by correlating each new data acquisition with the objects detected in the existing images. It was confirmed that the person tracking was performed with high accuracy even when the aerial video was subject to disturbances such as wind. However, this method lacks real-time performance because it is based on the assumption that human recognition is performed on videos recorded in advance from the air. In addition, because it is limited to the detection of persons, it cannot discriminate between behaviors. Therefore, we propose a real-time human action detection system from the air.

### 2.3. Aerial Human Violence Detection Using UAV

Srivastava et al. [11] have created a previously non-existent model of human violence recognition from the air. Violence recognition from drones can provide wide-area surveillance in a short time and improve security. Using video footage taken from an altitude of 2–8 m as a dataset, key points were extracted as classification model features to realize classification for each of the eight types of behavior (punching, kicking, falling, strangling, throwing, waving a pipe, and indicating SOS). Comparative validation of several classification models confirmed a detection accuracy of over 97% using SVM with an RBF kernel. However, violence detection is not performed in real-time, and detection needs to be performed after the video is captured. Therefore, it is not possible to take immediate action when violence is detected. Furthermore, the model can only be used in brightly lit situations. Therefore, we focus on detecting aerial behavior at night and use a unique model for thermal imaging cameras to realize real-time behavior detection at night.

### 2.4. Investigation of Comfortable Drone Human Approach Strategies

In the field of human–drone interaction, research has been conducted to discover the optimal strategy for drones to approach humans. Wojciechowska et al. [13] conducted indoor flight experiments with three parameters for each of four different items: distance, speed of approach, direction, and trajectory when the drone hovered near humans. Experimental results show that humans prefer a distance of 1.2 m when hovering, a speed of 0.5 m/s, a direction from the front, and an approach in a straight line for the trajectory. Comparing these results with experiments with ground robots, significant differences were confirmed. Kanda [14] showed that the distance preferred by humans to an interactive robot is about 40 cm. Walters et al. [15] indicated that when approached by a ground-moving robot, people preferred to approach from the right oblique front. Butler et al. [16] indicated that the robot's approach speed to a human at 0.25 m/s to 0.38 m/s was most preferred. These differences in the optimal approach methods between ground robots and drones are considered important in human–drone interaction. Our proposed drone security system needs to approach humans for projection when presenting information to people who are lost. However, no research has been conducted on drone proximity to humans at night. We conducted an experiment on the approach of drones to humans at night [17], based on the research method of Wojciechowska et al. As a result, a drone approach method at night that reduces fear and discomfort for humans was found.

## 3. Night Security Drone Aerial Ubiquitous Display

### 3.1. Aerial Ubiquitous Display Overview

Existing security drones have problems such as the difficulty of detecting human behavior at night and the lack of interaction functions such as providing information to people who have lost their way. This research solves both of these problems by developing an aerial ubiquitous display (AUD), a drone equipped with an infrared camera, and a projector. Figure 2 shows an overall view of the AUD and Table 1 shows the specifications of the AUD.

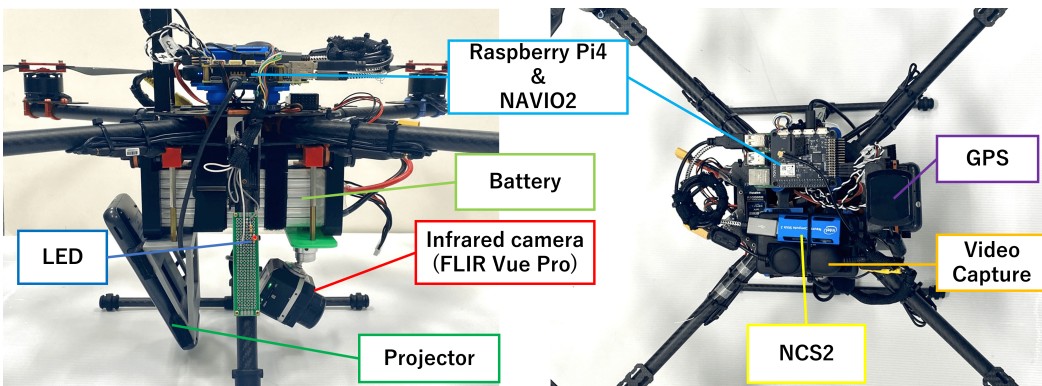

**Figure 2.** Aerial ubiquitous display (AUD).

**Table 1.** AUD specifications.

| | |
|---|---|
| Size (W, D, H) | 750, 750, 372 mm |
| Weight | 3650 g |
| The weight of a load (cameras, projectors, etc.) | 650 g |
| Payload (possible loadings other than main frame and battery) | 1500 g |
| Flight time | 20 min |
| Battery capacity | 8000 mAh |
| Voltage | 22.2 V |

*3.2. Hardware Configurations*

The TL65B01 manufactured by Tarot was used for the main body of the drone. The TL65B01 uses carbon for the frame, keeping the weight down to just 476 g. The drone has payload limitations, but the lightweight frame enables it to carry a variety of hardware. On top of the AUD, a Raspberry Pi4 and a NAVIO2 were mounted; the Raspberry Pi4 was used as a small computer to run the AUD, and the NAVIO2 as a flight controller. Raspberry Pi4 and NAVIO2 use UDP communication to send and receive information such as the aircraft's attitude and control commands. This allows the drone to fly according to the program executed on the Raspberry Pi4. The inference model for human detection was run on a Raspberry Pi4 mounted on the drone. Processing speed is an issue for real-time inference on a single-board computer such as the Raspberry Pi4. In the AUD, the Neural Compute Stick2 (NCS2) was installed as an item to improve the processing speed so that inference can be performed in real-time on the Raspberry Pi4. A FLIR Vue Pro thermal imaging camera was mounted on the bottom of the AUD to enable real-time detection of human activity, even at night. The FLIR Vue Pro was powered by a mobile battery. Video capture was also used, given that the FLIR Vue Pro has an analog video output. The video capture was connected to the Raspberry Pi4 via USB, allowing the use of the real-time video captured by the FLIR Vue Pro. Furthermore, a small projector was mounted on the bottom of the AUD to project information such as the current position and the distance to the destination to a person who is lost. Projecting information on the ground with a projector enables interaction between humans and drones, such as providing information to someone who calls for the AUD. This means that security is not limited to one-way camera-viewing of people, as is the case with existing security drones. When the AUD detects a waving behavior and decides that the person is lost and in trouble, the LED on the side of the AUD flashes. This LED serves to inform persons on the ground that the AUD has completed its decision.

### 3.3. Software Configuration

### 3.3.1. ArduPilot

AUD autonomous flight was implemented with ArduPilot [18], an open-source autopilot system. ArduPilot supports not only drones such as multicopters, but also various aircraft such as fixed-wing drones, boats, and submarines. In this study, ArduCopter [19], the firmware for multicopters within ArduPilot, was used because a quadcopter with four propellers was used. Copter3.6.9 was used for the ArduCopter version. Mission Planner [20] was used as the ground control station (GCS) supported by ArduPilot. The GCS can be used on a PC or mobile terminal to set up the aircraft and receive information such as position and flight path information during flight. In addition to these functions, Mission Planner can create flight path plans using waypoints. The routes can be easily created by pointing and clicking on the Google Map in the Mission Planner screen. In addition, the flight method at each waypoint can be set. This allows various types of movements to be set for each point, such as turning, hovering, decreasing altitude, and landing, as well as simply flying in sequence along the flight path.

### 3.3.2. Human Behaviour Detection Models

A thermal imaging camera, the FLIR Vue Pro, was used in the AUD to detect human behavior at night. To create a model for action detection using the FLIR Vue Pro, machine learning was required using a dataset of thermal imaging camera images taken from the air with the FLIR Vue Pro. In this study, a dataset was created using infrared images taken with the FLIR Vue Pro thermal imaging camera and an inference model for detecting human behavior using machine learning [21]. YoloV4-Tiny, a lightweight model of YoloV4 that performs fast and accurate object detection, was used as the model under which the action detection was performed. The Raspberry Pi in the AUD has 8 GB of memory, and the action detection model used in this study detects 0.8–1.2 frames per second. Figure 3 shows an example of a thermal image taken from the aerial that was used as the dataset. The model can identify two types of behavior in real-time: the existence of a human being on camera and hand waving. A black bounding box was displayed when a human was detected in the camera window and a blue bounding box when a waving person was detected. Figure 4 shows an example of detecting the existence of a human being and a hand-waving motion. This model is run on a Raspberry Pi4 mounted on an AUD during night security patrols to achieve aerial action detection.

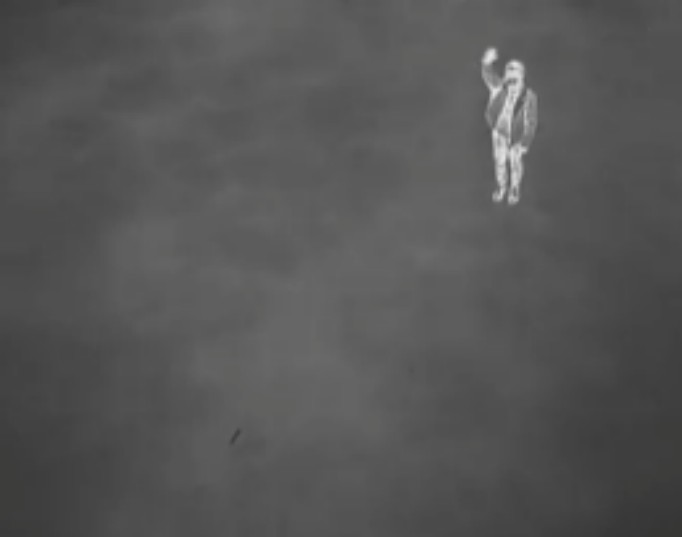

**Figure 3.** Examples of aerial images used as data sets.

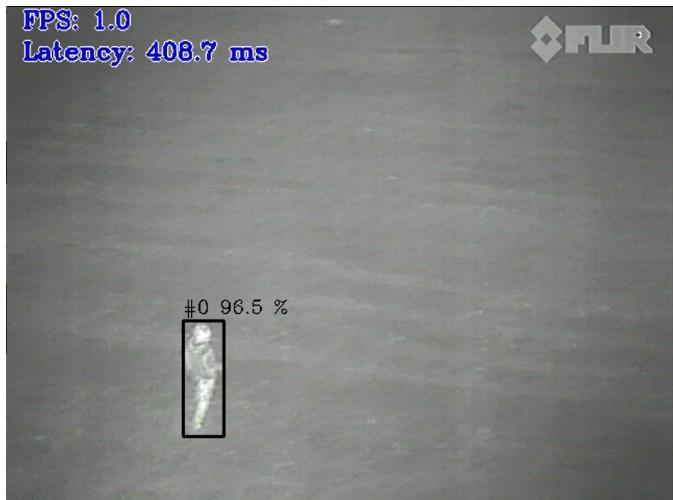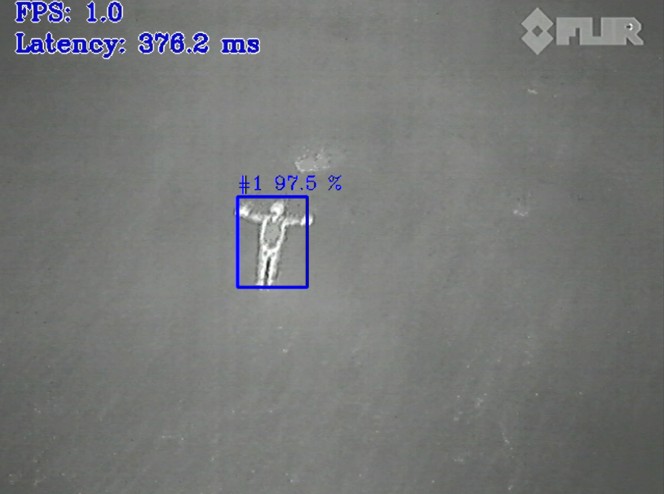

**Figure 4.** Examples of real-time aerial action detection. (**Left side**) A black bounding box indicates human presence. (**Right side**) The blue bounding box indicates the movement of the human hand waving.

### 3.3.3. DroneKit

When an AUD detects a human waving a hand to request information to be displayed, it is required to decrease altitude in order to project the information on the ground. When a human presence is detected at night, the AUD is required to maintain hovering at the detected position in order to monitor the person and ensure that the person is not suspicious. In this study, the control of the AUD in each of these situations was enabled by using the DroneKit [22]. DroneKit enables the acquisition of the status and various parameters of the connected aircraft and the control of the flight controller by programs. The Python API [23] within DroneKit was used in the AUD. The Python program can be run on the Raspberry Pi4 mounted on the AUD to control the aircraft by flying autonomously to a specified latitude and longitude, hovering, changing altitude, and so on. Socket communication between the human behavior detection model and the DroneKit program enables real-time and automatic control of the AUD according to the detection results, such as decreasing the altitude for information projection when a waving human is detected, or hovering for monitoring when a human existence is detected.

## 4. Nighttime Security System with Aerial Ubiquitous Display

### 4.1. System Overview

In this study, a drone AUD equipped with an infrared camera and a projector is used to provide night security. The following functions are required for the AUD to patrol at night, presenting information to people who have lost their way and monitoring people who may be suspicious.

1. Patrol flights on security routes.
2. Real-time human behavior detection and alert function at the time of detection.
3. Hovering for monitoring when detecting human presence.
4. Decreasing altitude and hovering for information projection when detecting hand-waving behavior.
5. Projecting information using a projector.

Figure 5 shows a flowchart of the flow of nighttime security operations performed by the AUD.

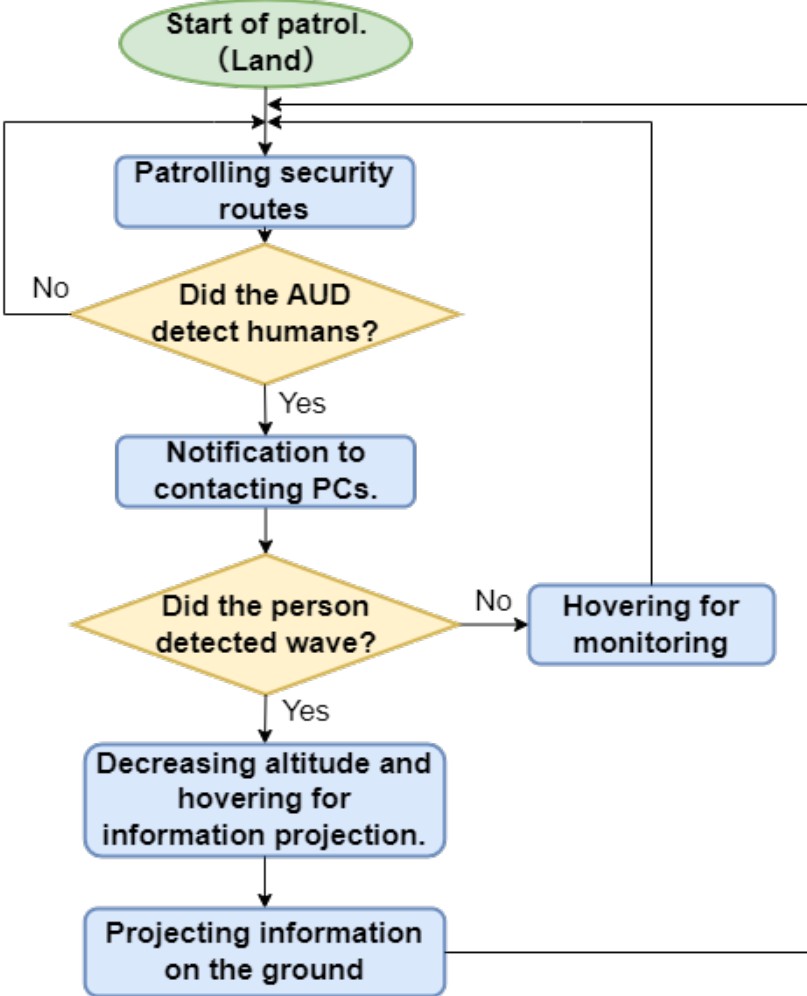

**Figure 5.** Flowchart of night security by the AUD.

*4.2. Functions in the System*

This section provides a description of the five functions required nighttime security system with AUD.

### 4.2.1. Patrol Flights on Security Routes

In order for the AUD to perform security operations alone, it needs to fly autonomously along a pre-defined security route. Mission Planner, a GCS supported by ArduPilot, was used to create the security route. Waypoints were created in advance using Mission Planner's Create Flight Plan function. The flight plan specifying speed, altitude, etc., was saved on the Raspberry Pi4 mounted on the AUD. This enabled the patrol to fly along the pre-created security route. When AUDs fly patrol routes at night, there is a requirement to reduce the flight noise from the AUD to which people on the ground are exposed and to detect human activity with high accuracy using thermal imaging cameras. In this study, several altitudes were prepared as patrol altitudes for AUDs, and experiments were conducted at night to find the most suitable altitude from two points of view: the investigation of AUD flight noise heard on the ground and the accuracy of action detection from the air. Examples of the results of action detection at each altitude are shown in Figure 6. Experimental results confirmed that human behavior is most accurately detected at an altitude of 10 m and that the AUD flight noise heard on the ground decreases with altitude, but the difference decreases at higher altitudes. Table 2 shows the accuracy of the behavior detection and the sound level of the AUD flight noise at different altitudes

obtained from the experiments. Based on these results, the patrol altitude for the AUD security at night was decided to be 10 m.

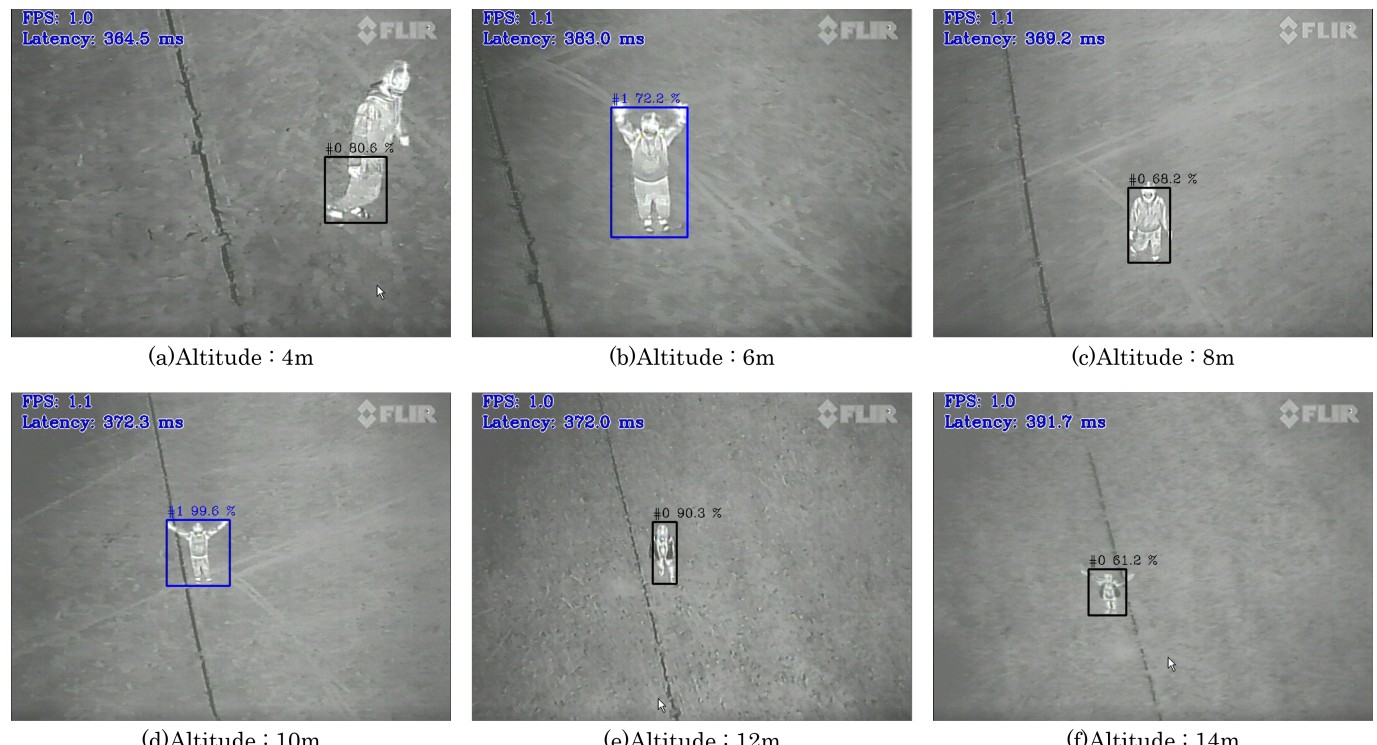

(a)Altitude：4m     (b)Altitude：6m     (c)Altitude：8m

(d)Altitude：10m     (e)Altitude：12m     (f)Altitude：14m

**Figure 6.** Example of behavior detection results when the hovering altitude of the AUD is varied every 2 m.

**Table 2.** Results of action detection at different altitudes and the volume of sound heard from the ground at different altitudes.

| Altitude (m) | TP | FP | FN | TN | Precision | Recall | F-Measure | Sound Level (dB) |
|---|---|---|---|---|---|---|---|---|
| 4 | 0 | 6 | 11 | 19 | 0 | 0 | 0 | 73.1 |
| 6 | 20 | 9 | 3 | 4 | 0.69 | 0.87 | 0.770 | 72.2 |
| 8 | 24 | 8 | 4 | 6 | 0.75 | 0.86 | 0.801 | 70.6 |
| 10 | 39 | 8 | 1 | 5 | 0.83 | 0.98 | 0.899 | 68.2 |
| 12 | 30 | 7 | 3 | 3 | 0.81 | 0.91 | 0.857 | 66.3 |
| 14 | 22 | 8 | 7 | 2 | 0.73 | 0.76 | 0.745 | 66.4 |

4.2.2. Real-Time Human Behavior Detection and Alert Function at the Time of Detection

For the action detection model in this study, a model created in a previous study for FLIR Vue Pro was used [21]. This model was performed on a Raspberry Pi4 mounted on a drone to enable real-time action detection from the air. When an action is detected, a notification sounds on the PC communicating with the AUD, so that even if the operator is away from the camera image, when they hear the notification sound, they will know that the AUD has detected a human. In this study, the number of detections was counted for both human existence and hand-waving behavior, and four frames of detection were determined as the requirement for determining the behavior. When four frames of human presence (the black bounding box) were detected, the system shifted to hovering for surveillance, and when four frames of waving behavior (the blue bounding box) were detected, the system shifted to altitude reduction and hovering for information projection. When four frames were detected in either direction, the counts for both the human presence and the waving behavior were reset. This technique was used to prevent hovering and loss

of altitude in unexpected positions in the case of false detection of non-human objects. Furthermore, to prevent repeated monitoring or information projection on the same person, the number of detections was not counted during monitoring or information projection. Figure 7 shows an example to determine the existence of a human and an example to determine hand-waving behavior.

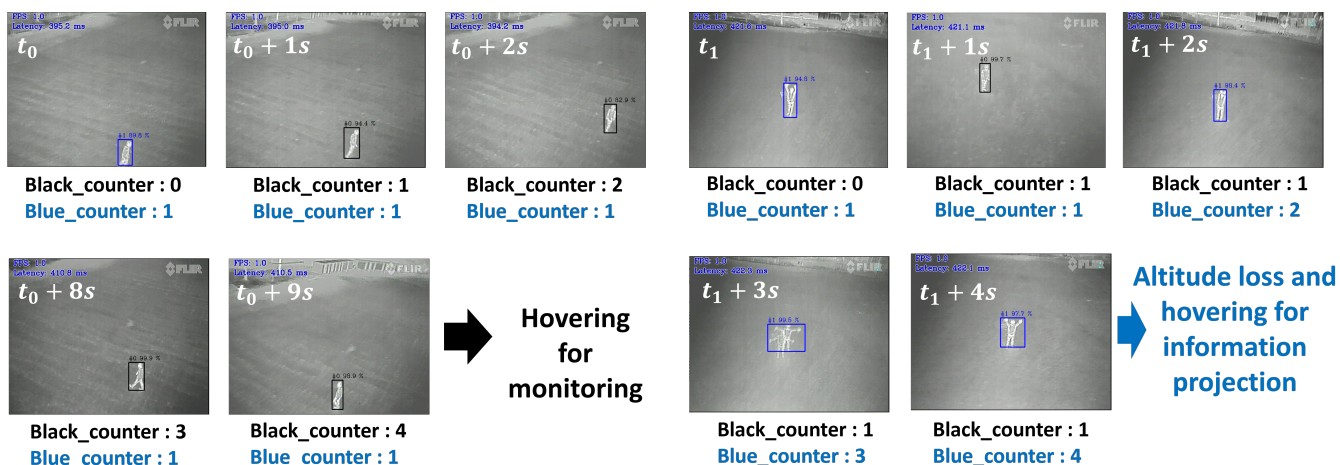

**Figure 7.** Flow of determining the presence of a human and hand-waving behavior. The five images on the right ($t_0 \sim t_0 + 9s$) are determining the presence of a human and the five images on the left ($t_1 \sim t_1 + 4s$) are determining the hand-waving behaviour.

### 4.2.3. Hovering for Monitoring When Detecting Human Presence

The AUD was developed for use in late-night campus patrols. If a person is present on campus late at night, they may be suspicious. If AUD detects the presence of a person during night patrols and decides that the person is a possible suspicious person (detection of the presence of a person in four frames), it will hover on the spot for a certain period of time to monitor the person's movements. When a human is detected, a notification sound is sent on the PC communicating with the AUD. This enables the administrator to check what the detected person is doing and what the situation is through the camera image on the PC. In addition, a 10 m altitude, similar to the patrol altitude, was used for the altitude because it is considered preferable to be able to monitor a wide area when monitoring.

### 4.2.4. Decreasing Altitude and Hovering for Information Projection When Detecting Hand-Waving Behavior

In the night security system with AUD, the AUD hovers at a reduced altitude and project information to people who need information such as their current position and distance to their destination, such as people who have lost their way. When the AUD detects waving behavior and decides that the person is lost (detection of waving behavior in four frames), it loses altitude on the spot and projects information on the ground. Hovering during information projection requires an altitude at which fear and discomfort are low and information is easy to read. In this study, an experiment was conducted to investigate the most suitable hovering altitude for AUD to present information with a projector. Information was projected from the air to nine subjects at two different hovering altitudes (2 m and 4 m) and evaluated using a questionnaire and self-assessment manikin (SAM) [24]. Figure 8 shows the scene during the experiment. The experimental results confirmed that projecting information at an altitude of 2 m was less uncomfortable and frightening, and the projected information was easier to read. Based on these results, it was decided that the hovering of the AUD information projection should be done at an altitude of 2 m.

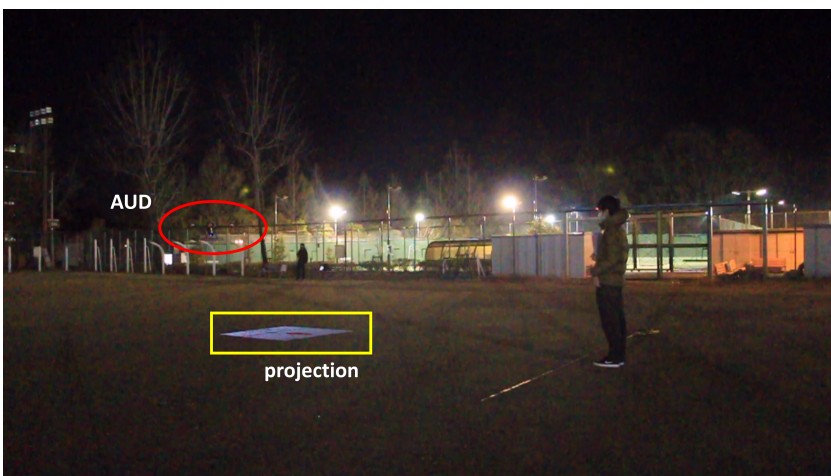

**Figure 8.** The AUD projection experiment.

4.2.5. Projecting Information Using a Projector

The presentation of information to people who need information at night was achieved by projecting the information from a projector mounted on the AUD to the ground. The information to be projected includes the current position and the direction and distance to the destination. This projection enables a person who is lost to know their current position, nearby buildings, etc. Changing the projection content depending on the location was achieved by using the waypoint numbers of the patrol route. When the waving behavior was detected, the waypoint number was obtained and the image corresponding to that number was displayed. The combination of the projector and the thermal imaging camera thus enables interaction between the human and the drone and not only camera-based monitoring as in the case of existing security drones.

## 5. Experiment

### 5.1. Experimental Details

Experiments were conducted to simulate an actual security scenario using the AUD and were developed with the concept of performing night security on campus in place of security guards. The purpose of this experiment was to assess whether the AUD was functioning satisfactorily as a night security drone. The experiment was conducted on the large grounds of the Biwako-Kusatsu campus of Ritsumeikan University. The subjects were six Ritsumeikan University students aged between 21 and 23. Flights during the experiment were performed along pre-designed waypoints to simulate real nighttime security. Additionally, the experiment was conducted at night, when it was sufficiently dark. Figure 9 shows the flight paths of the experiment on the ground. The points shown in Figure 9 are the points through which the AUD passes during the flight. It takes off from the position marked "H" and lands at point "9". At each point, the AUD hovered for 10 s while maintaining a patrol altitude of 10 m. This 10 s of hovering was set up to enable the AUD to detect and decide on human behavior. In this experiment, three different scenarios were conducted using the flight path shown in Figure 9, and a total of six patrol flights were conducted, two for each scenario, to test the night security. Two different roles were played by all the subjects, one as a possible suspicious person and the other as a person lost at night, during the six flights. The three scenarios prepared are described in detail in Sections 5.1.1–5.1.3.

5.1.1. Person Monitoring Scenario

It is considered that sufficient monitoring functions are required for a security drone to perform security patrols. In this scenario, a task was evaluated in which the AUD detects and monitors the presence of humans during night security patrols. Subjects played the role of a possible suspicious person, walking or stopping around the waypoints of the flight

path shown in Figure 9. For each flight experiment, three or four subjects were placed in random positions around the waypoint. When the AUD detected four frames of human presence, it determined the presence of a possible suspicious person and monitored it by hovering at an altitude for 10 s, after which it returned to its patrol path.

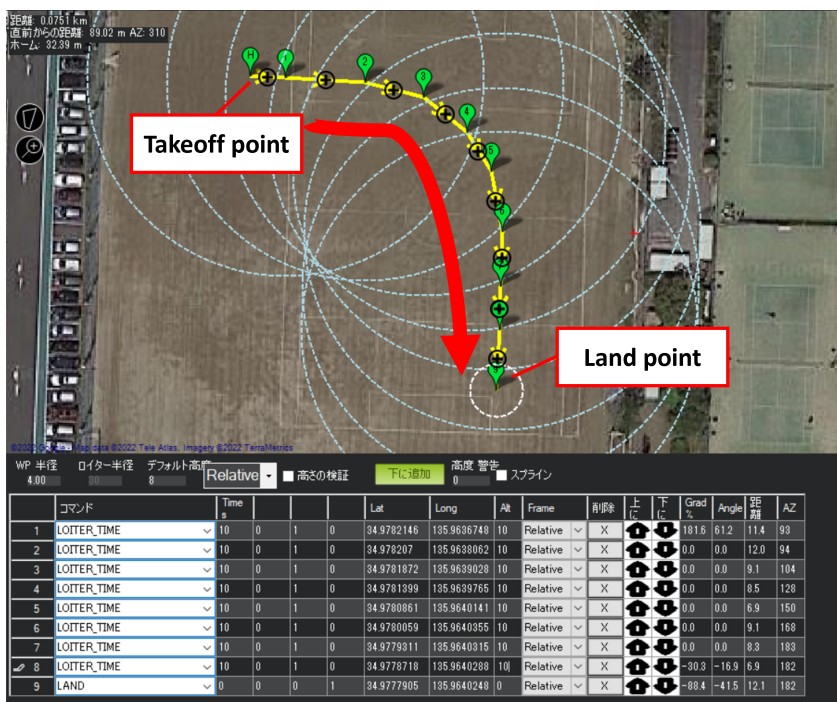

**Figure 9.** Flight path in the experiment on the ground of the Biwako-Kusatsu Campus of Ritsumeikan University.

### 5.1.2. Information Projection Scenario for Waving Person

The AUD not only monitors the area with cameras but also has the function of displaying information to persons who are waving their hands. This has enabled interactions such as providing information on buildings in the surrounding area to people who have lost their way. In this scenario, the AUD was evaluated for the task of detecting the behavior of a waving person during a night security patrol and displaying information to that person. The subject played the role of a person lost at night and waved his hands at the AUD on the flight path of the waypoint shown in Figure 9. For each flight experiment, three or four subjects were placed in random positions around the waypoint. When the AUD detects four frames of a waving person's behavior, it determines that the person is lost and seeking information, decreases the altitude to 2 m, and projects the information using a projector. The three pieces of information projected were the name, direction, and distance of nearby buildings on the university campus where the experiment was conducted. During the projection, the AUD maintains an altitude of 2 m, hovering for 10 s. After the information projection, the altitude is increased to 10 m and the AUD returns to the patrol path. Additionally, the subject playing the role of a lost person receives information by seeing the information projected on the ground by the projector while the AUD is hovering at an altitude of 2 m.

### 5.1.3. Mixed Scenario of Monitoring and Information Projection

Night security using the AUD required the use of two different types of movement, depending on the actions of the person on the ground during the patrol, to monitor for possible suspicious persons and to provide information to people who are lost and waving their hands. Therefore, in this scenario, two different scenarios, Sections 1 and 2, were mixed and evaluated to test whether the AUD was able to respond to each person's movements.

Four subjects were placed in random positions around the waypoint path on each flight of the experiment, with two playing the role of a potentially suspicious person and the other two playing the role of a person who is lost and seeking information. When four frames of a person walking or stopping are detected, the system determines that the person may be suspicious and monitors the area for 10 s. When four frames of a person waving their hands are detected, the system determines that the person is lost and decreases the altitude from 10 m to 2 m, and projects the information for 10 s. The information projected here is the same as that projected in Section 5.1.2.

*5.2. Evaluation Method*

The evaluation of this experiment was divided into two sections.

1. Evaluation of the accuracy of decisions in action detection.
2. Evaluation of monitoring and information projection through questionnaires.

The accuracy of the decision to detect during a patrol flight was evaluated based on how accurately the AUD was able to determine the behavior of the subjects who played the two roles in the experiment. In this experiment, a total of 22 decision opportunities were provided in the path shown in Figure 9: 11 opportunities to decide on the role of a possible suspicious person and 11 opportunities to decide on the role of a person waving their hands in a request for information. The actual number of correct decisions made on these 22 opportunities was evaluated and discussed. Monitoring and information projection questionnaire assessments were administered to all subjects at the end of the complete experiment. Subjects who played two different roles answered each question with the choice of feeling that was closest to what they felt from a list of five options. At the end of the questionnaire, a free text box was provided, in which the subjects could write down their feelings and impressions throughout the experiment. Table 3 shows the questionnaire questions and Table 4 shows the options for the questions.

**Table 3.** Questionnaire questions used in the experiment.

| No. | Question Content |
|---|---|
| I | When playing the role of the suspicious person, did you feel that you were being monitored by the drone? |
| II | Did you feel that the AUD would be useful as a robot to monitor potentially suspicious persons? |
| III | How did you feel about the time required for the drone to wave its hands before it could project information? (Only subjects who successfully determined the waving behavior in the role of a lost person answered). |
| IV | Did you feel that the AUD would be useful as a robot to present information to people who are lost? |

**Table 4.** Questionnaire options used in the experiment.

| No. | Options |
|---|---|
| I | 1. None at all 2. Not much 3. Neither 4. A little 5. Very much |
| II | 1. Not useful at all 2. Not very useful 3. Neither 4. Quite useful 5. Very useful |
| III | 1. Very long 2. A little long 3. Just right 4. A little short 5. Very short |
| IV | 1. Not useful at all 2. Not very useful 3. Neither 4. Quite useful 5. Very useful |

## 6. Results

The evaluation of decision accuracy was calculated according to the success or failure of the decision on human detection occasions during the experiment. Table 5 shows a list of the number of successes, failures, and success rates of decisions made during action detection for each scenario in a total of six flight experiments, while Table 6 shows the sum of the decision results from all the experiments. In the questionnaire, the responses of all subjects were counted, graphed, and then discussed. The results obtained from the counting of each question in the questionnaire are shown in Figure 10.

**Table 5.** Decision results for action detection in each scenario.

| | Determining Potential Suspicious Persons | | | Determining Who Lost Their Way | | |
|---|---|---|---|---|---|---|
| | **Success** | **Failure** | **Success Rate** | **Success** | **Failure** | **Success Rate** |
| Monitoring scenario 1 | 2 | 1 | 66.7% (2/3) | - | - | - |
| Monitoring scenario 2 | 2 | 2 | 50.0% (2/4) | - | - | - |
| Projection scenario 1 | - | - | - | 3 | 0 | 100% (3/3) |
| Projection scenario 2 | - | - | - | 3 | 1 | 75.0% (3/4) |
| Mixed scenario 1 | 2 | 0 | 100% (2/2) | 2 | 0 | 100% (2/2) |
| Mixed scenario 2 | 1 | 1 | 50.0% (1/2) | 1 | 1 | 50.0% (1/2) |

**Table 6.** Sum of judgment results of all experiments.

| | **Success** | **Failure** | **Success Rate** |
|---|---|---|---|
| Determining potential suspicious persons | 7 | 4 | 63.6% (7/11) |
| Determining who lost their way | 9 | 2 | 81.8% (9/11) |
| Sum of decisions on possible suspicious persons and lost persons | 16 | 6 | 72.7% (16/22) |

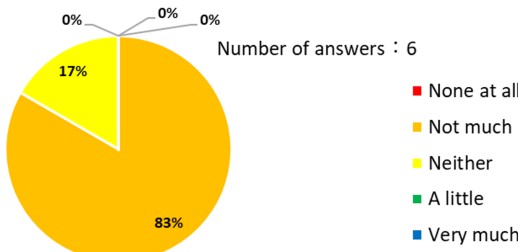

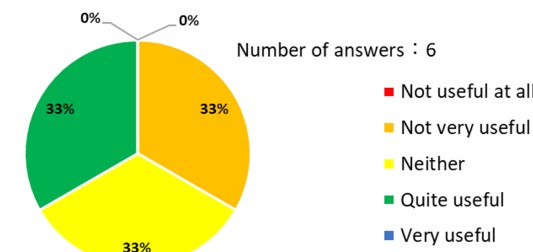

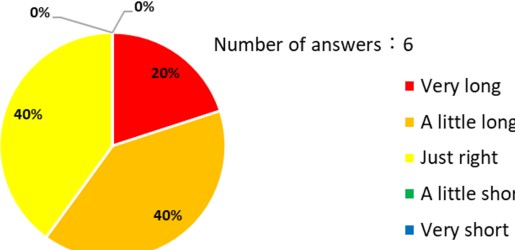

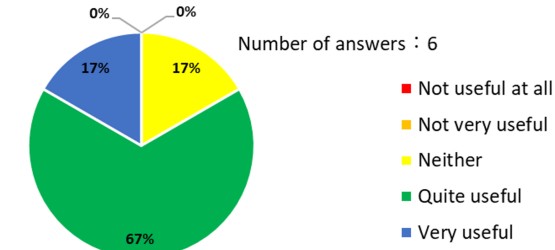

**Figure 10.** Results of the questionnaire. The top left graph shows question number Ⅰ, the top right graph shows question number Ⅱ, the bottom left graph shows question number Ⅲ, and the bottom right graph shows question number Ⅳ.

## 7. Discussion

### 7.1. Discussion of Monitoring for Possible Suspicious Persons

In this experiment, when the drone determined that a person walking or standing still during its patrol was a potentially suspicious person, it hovered for 10 s, during which time the drone monitored the person from a camera. The same method as in Figure 7 was used for the decision method, and only when the presence of a human being was detected for more than four frames, it was determined that the person was a possible suspect. As a result, AUD successfully monitored 7 out of a total of 11 opportunities for decisions,

resulting in a success rate of 63.6% for the decision to identify a possible suspicious person. Figure 11 shows an example of the camera image of a successful decision and the scene of the experiment, and Figure 12 shows an example of the camera image of a failed decision. As shown in Figure 1, when a person walking or standing still was successfully determined as a possible suspicious person, the AUD hovered for 10 s as expected and captured the subject on camera. On the other hand, none of the four failed cases were monitored, and they continued patrolling the experimental route past the subject. Two problems were considered to be the causes of this situation. The first problem is that humans are not detected accurately. As shown in Figure 12, there were several frames in which a person walking or standing still in the camera image was not detected as a human being. Therefore, it did not detect the presence of a human being for the four frames required to determine that the person was a possible suspicious person during the 10 s of hovering at the waypoint. The second problem is that no humans were seen in the camera images. Due to the environment during the experiment, the posture of the AUD was not stable when hovering, and there were scenes in which the camera looked at positions outside the patrol path of the experiment. As a result, the AUD was unable to detect humans and failed to determine who was possibly suspicious. In the questionnaire from the subjects, as shown in Figure 10, the majority of the subjects answered that they did not feel much like being monitored when asked if they had a sense of being monitored, and in the evaluation of the usefulness of the monitoring using the AUD, all the subjects chose between not very useful and quite useful. In the comments section, we received comments such as "I did not feel that I was being monitored because the drone did not intentionally change its posture to look at me", and "I did not feel anything because there were no warnings from the drone". Based on these results, it was considered necessary to not only monitor the person on the spot but also to follow the person or warn the person using sound or light when the system decides that the person is a possible suspicious person in order to suppress suspicious behavior. In addition, it was considered necessary to improve the accuracy of the model to increase the accuracy of the decision.

### 7.2. Discussion of Information Projection to a Waving Person

In this experiment, when the AUD detected the waving behavior while patrolling and determined that the person was lost, it decreased its altitude to 2 m, hovered for 10 s, and projected information on the ground with a projector. The same as in Figure 7 was used for the decision method, and only when four or more frames of waving behavior were detected, the person was decided to be a person who was lost and seeking information. As a result, the AUD succeeded in projecting information 9 times out of a total of 11 opportunities to decide, and the success rate for the decision of the person who got lost and waved was 81.8%. Figure 13 shows an example of the camera image of a successful decision and the scene of the experiment, and Figure 14 shows an example of the camera image of a failed decision. As shown in Figure 13, when the AUD successfully decided that a person waving toward it was an information seeker, it decreased its altitude to 2 m, hovered for 10 s, and projected information toward the ground, as expected. Furthermore, all subjects who made successful decisions were able to read the content of the information projection on the ground. On the other hand, the two failed cases did not lose altitude and no information projection was made. These two failures were due to the presence of multiple frames, as shown in Figure 14, in which a waving motion was falsely detected as the presence of a human or the camera did not detect the action of the human in the camera. In one case, the false detection of human behavior caused the drone to perform monitoring on a human who should have been performing the information projection. Furthermore, through the questionnaire, it was confirmed that many subjects felt that the time required for the drone to wave its hands before projecting information was too long (Figure 10). Based on this, in this experiment, the number of frames of detection of the hand-waving behavior necessary to decide who is requesting information was set at four frames, but it was considered necessary to make the decision using a smaller number of frames. In order

to make decisions in fewer frames, more accurate action detection models will be needed to avoid projecting information at unexpected points. As shown in Figure 10, Question IV, which evaluates the usefulness of information presentation using an AUD, was answered as quite useful by many subjects, and one subject answered it as very useful. This is thought to be due to the higher success rate obtained compared to the decision of the suspicious person and the fact that the subjects were able to read the projected information accurately. In the comment section, we received comments such as "I found it useful and informative to read the information". The results suggest that projecting information using a projector to a person lost in the AUD is effective for night security using a drone. However, it is necessary to improve the method of deciding the hand-waving behavior to provide an interaction that is less stressful for the user.

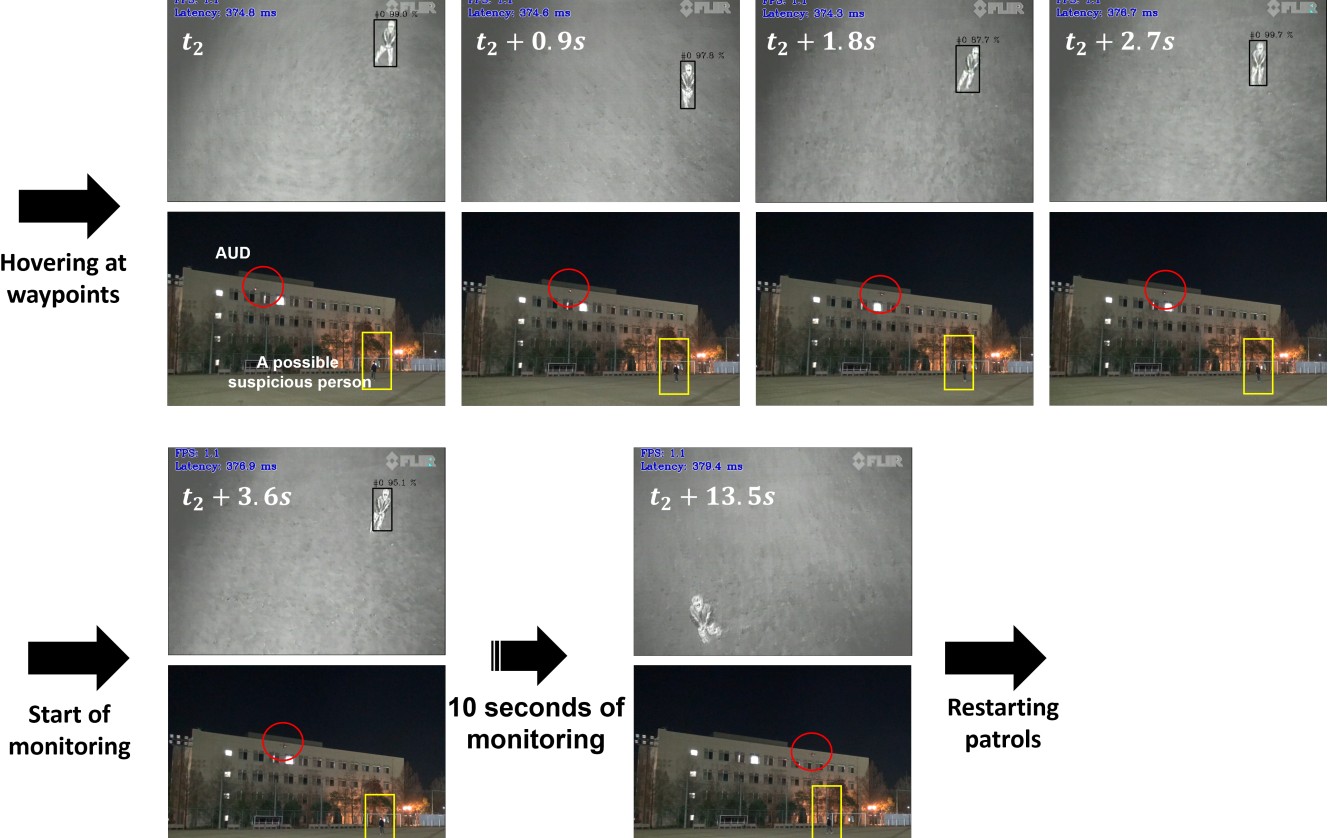

**Figure 11.** An example of the AUD detecting 4 frames of human presence and monitoring a person as a possible suspicious person. The upper row shows the camera image and the lower row shows the scene during the experiment.

### 7.3. Discussion of All Scenarios

The success rate of all decisions made in this experiment was 72.7%, and 16 decisions were made successfully out of 22 opportunities. The success rate did not decrease significantly in the mixed scenario of monitoring and projection, and the AUD was able to make two types of decisions during a single patrol and perform movements according to the results of each decision. The reasons for decision failures in all scenarios were the failure of the camera to capture the target person and the false detection of the actions. For the point where the camera could not capture the human, it was considered necessary to reduce the blind spots by changing the posture of the AUD. Further improvement of the action detection model is needed to reduce false detections. Furthermore, it was thought that a smoother decision could be achieved if the frame rate of action detection could be further improved. In this experiment, we were not able to achieve a high success rate of over

90%. However, we were able to develop an unprecedented security method for real-time monitoring and information projection during night patrols on actual flights.

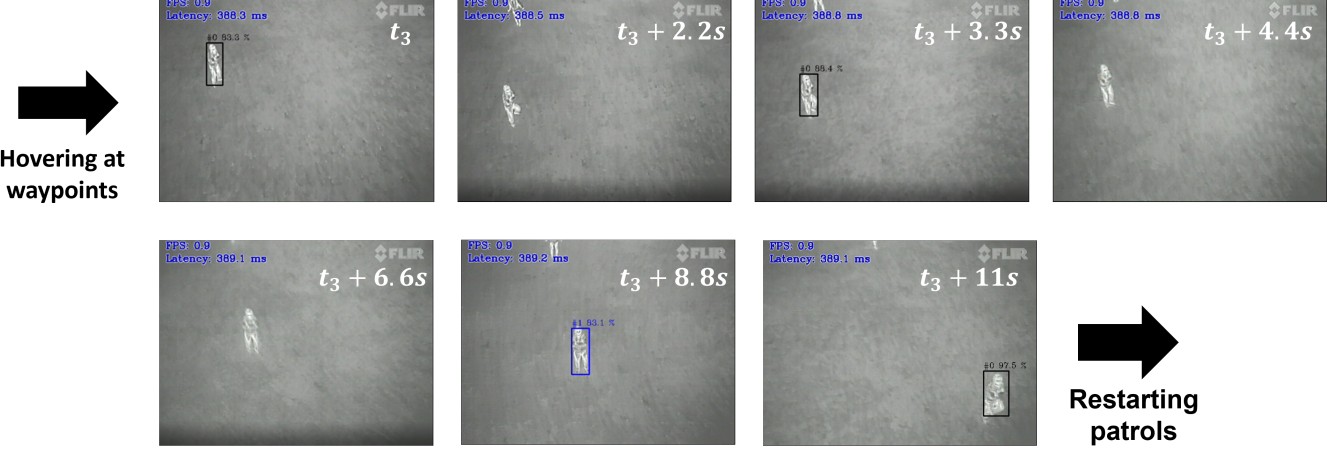

**Figure 12.** An example camera image of a failed decision of a possible suspicious person.

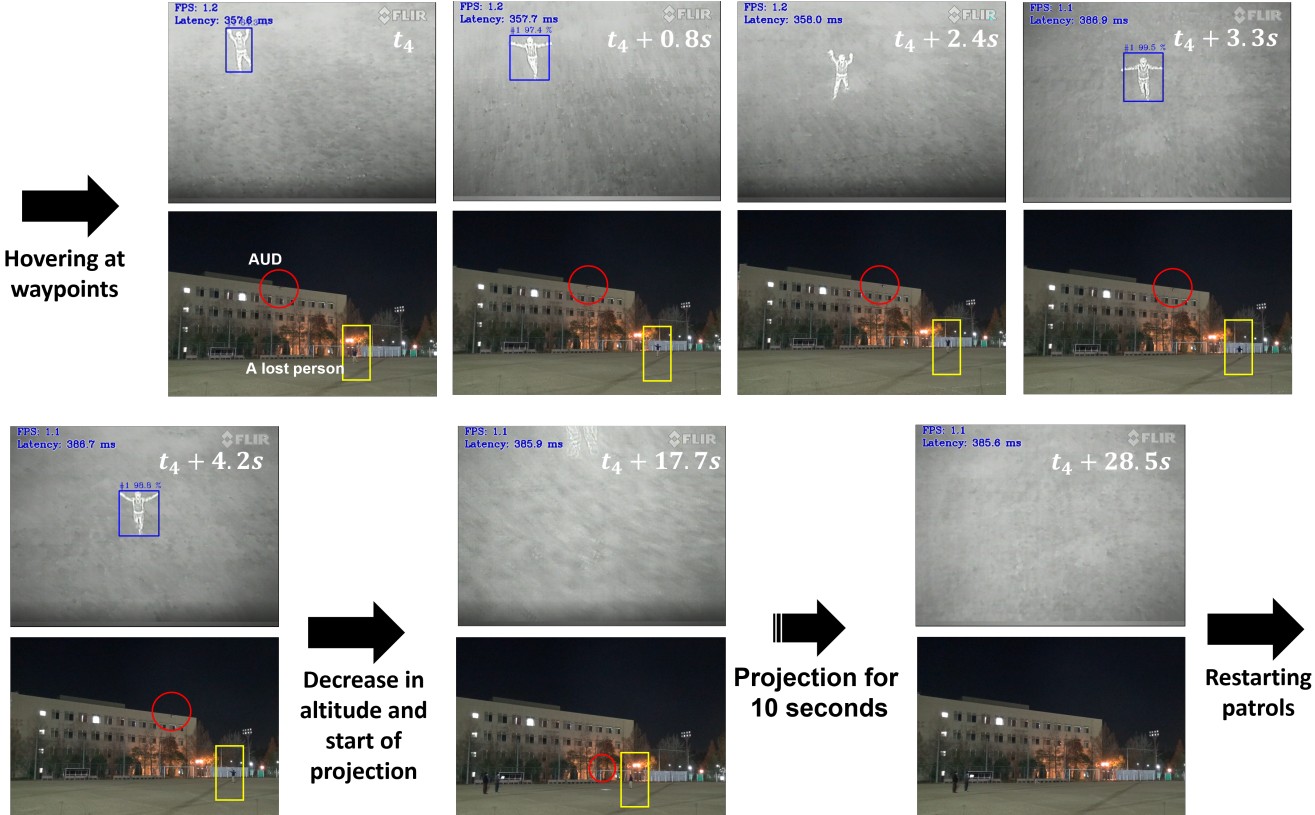

**Figure 13.** An example of the AUD detecting 4 frames of waving behavior and projecting information to a person who has lost their way. The upper row shows the camera image and the lower row shows the scene during the experiment.

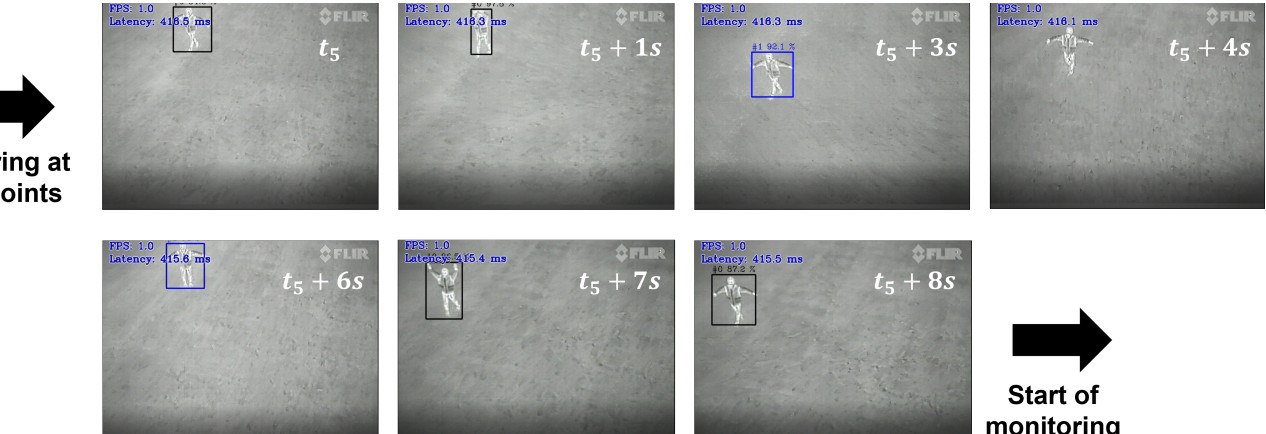

**Figure 14.** An example camera image of a failed information projection.

## 8. Conclusions

In this paper, an AUD developed as a night security drone was proposed to reduce the labor shortage and physical burden of security guards, which have become problems in recent years. A system was also proposed to provide night security using the AUD. In this study, experiments were conducted using an AUD to simulate actual patrol security. Three scenarios were prepared, and the movements of the AUD were evaluated in each of the possible situations. The results showed that the success rate of monitoring a potentially suspicious person was 63.6% and the success rate of projecting information to a lost person was 81.8%, resulting in an overall success rate of 72.7%. The information projection interaction using the AUD was shown to be useful for night security. Through this experiment, it was shown that real-time night security from the air is possible using the AUD. In the future, it is necessary to use more accurate action detection models to find a decision method with fewer failures. The altitude limit of behavior detection can be improved by using models that can handle a wider variety of situations, and it is expected that accurate detection will be possible at altitudes below 10 m as well as at higher altitudes. At this stage, only 0.8 to 1.2 frames per second are detected, but as this number increases, more advanced decisions can be made. It was also considered necessary to consider techniques for self-localization estimation that do not depend on satellites such as GPS, with the aim of actually guarding not only the ground but also the areas around buildings.

**Author Contributions:** Methodology, R.K.; Software, R.K.; Validation, R.K.; Formal analysis, R.K.; Writing—original draft, R.K.; Writing—review & editing, R.K., D.T.T. and J.-H.L.; Supervision, D.T.T. and J.-H.L.; Project administration, D.T.T. and J-H.L. All authors have read and agreed to the published version of the manuscript.

**Funding:** This research received no external funding.

**Informed Consent Statement:** Informed consent was obtained from all subjects involved in the study.

**Data Availability Statement:** Not applicable.

**Conflicts of Interest:** The authors declare no conflict of interest.

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
