# Peer review of "Evaluation of Human Behaviour Detection and Interaction with Information Projection for Drone-Based Night-Time Security"

_drones, doi:10.3390/drones7050307_

Round 1

Reviewer 1 Report

Great work addressing prior comments. I have one suggestion to improve the abstract, where the authors pointed out that current methods are limited by range, but the proposed method's limitation seems to be the range of application (4.5 m, 5 Hz) therefore I would not outline in the abstract that range is one of the limitations that the proposed solution is offering. I would also add at the beginning (and probably at the end too) that this is the first step towards the use of innovative displacement sensing that later can overcome distance and frequency limitations. I think it is very innovative and the limitations are not a problem, but it is best to explain that in the future they will go away for actual applications.

Reviewer 2 Report

This is an interesting paper but I think it can be greatly improved if the authors explained better the machine learning algorithm, how it was actually implemented on the Raspy, what is the memory footprint and time needed to compute the solution.

In the beginning of the paper the authors talk about indoor and outdoor environments but the experiments are performed only on outdoor environments. 

The projecting scenario is not clear. I would like more information on this   issue. The way in which the projection is done, what is the actual information projected.

There is no evaluation or comparison to a set of fixed cameras vs the drone approach. While the drone may patrol using a periodic pattern, fixed surveillance cameras will do that in a continuous way. 

If cameras are correctly deployed they can detect the movement of people in the different areas while the intruder moves along his/her path. The drone may only detect he/she on a fixed spot and eventually the intruder may hide from the drone.

The altitude issue should be better explained. 10 m is quite a lot and that would not be possible to use in indoor environments, for sure.

Reviewer 3 Report

The overall research idea to replace the security guard with drones is novel and can significantly impact drone application areas.  To improve the quality of this paper, some minor revisions can be made:

 1. Raspberry Pi integrates with an infrared camera based on computer vision.  What is not clear is which detection algorithm is used in this research, i.e., open cv, Yolo, etc.

 2. Raspberry Pi doesn't have a fast response for image processing. It is better to include extra information about the response time to detect humans.  Is it possible to detect when someone runs or moves fast?  It can't be detected if the response time is slow and the drone moves quickly.  The limitation of human speed as a detected object can be added

 3. Suspicious person is determined when they are walking or standing still for a while. If the focus is on security, this sentence is not quite right. It is more apt to limit the observation to a restricted area where the situation may happen.

 4. The hardware and software configuration ardupilot is used, but the communication between the flight controller and raspberry pi is not visible.  What is the data transferred between these two systems?

 5. In the concluding section, adding the detection distance to the system and the camera's field of view is better.

 Hopefully, my comments can improve your work.
